# Sensitivity of the Natriuretic Peptide/cGMP System to Hyperammonaemia in Rat C6 Glioma Cells and GPNT Brain Endothelial Cells

**DOI:** 10.3390/cells10020398

**Published:** 2021-02-15

**Authors:** Jacob T. Regan, Samantha M. Mirczuk, Christopher J. Scudder, Emily Stacey, Sabah Khan, Michael Worwood, Torinn Powles, J. Sebastian Dennis-Beron, Matthew Ginley-Hidinger, Imelda M. McGonnell, Holger A. Volk, Rhiannon Strickland, Michael S. Tivers, Charlotte Lawson, Victoria J. Lipscomb, Robert C. Fowkes

**Affiliations:** 1Endocrine Signalling Group, Department of Comparative Biomedical Sciences, The Royal Veterinary College, University of London, Royal College Street, London NW1 0TU, UK; jacobtcregan@icloud.com (J.T.R.); samantha.byers@admin.cam.ac.uk (S.M.M.); cscudder@rvc.ac.uk (C.J.S.); emily.stacey@hotmail.co.uk (E.S.); sabahkhan35@gmail.com (S.K.); worwoodmike@gmail.com (M.W.); tp00117@surrey.ac.uk (T.P.); sebdennisberon@gmail.com (J.S.D.-B.); mginleyh@gmail.com (M.G.-H.); 2Department of Comparative Biomedical Sciences, The Royal Veterinary College, University of London, Royal College Street, London NW1 0TU, UK; imcgonnell@rvc.ac.uk (I.M.M.); chlawson@rvc.ac.uk (C.L.); 3Stiftung Tierärztliche Hochschule Hannover, Klinik für Kleintiere, Bünteweg, 930559 Hannover, Germany; holger.volk@tiho-hannover.de; 4Clinical Sciences & Services, Hawkshead Lane, North Mymms, Hatfield, Hertfordshire AL9 7TA, UK; rstrickland@rvc.ac.uk (R.S.); vlipscomb@rvc.ac.uk (V.J.L.); 5Paragon Veterinary Referrals, Paragon Business Village Paragon Way, Red Hall Cres, Wakefield WF1 2DF, UK; mtivers@alumni.rvc.ac.uk

**Keywords:** natriuretic peptides, cGMP, hyperammonaemia, astrocyte, neuroendocrinology, endothelial cells, extracellular vesicles

## Abstract

C-type natriuretic peptide (CNP) is the major natriuretic peptide of the central nervous system and acts via its selective guanylyl cyclase-B (GC-B) receptor to regulate cGMP production in neurons, astrocytes and endothelial cells. CNP is implicated in the regulation of neurogenesis, axonal bifurcation, as well as learning and memory. Several neurological disorders result in toxic concentrations of ammonia (hyperammonaemia), which can adversely affect astrocyte function. However, the relationship between CNP and hyperammonaemia is poorly understood. Here, we examine the molecular and pharmacological control of CNP in rat C6 glioma cells and rat GPNT brain endothelial cells, under conditions of hyperammonaemia. Concentration-dependent inhibition of C6 glioma cell proliferation by hyperammonaemia was unaffected by CNP co-treatment. Furthermore, hyperammonaemia pre-treatment (for 1 h and 24 h) caused a significant inhibition in subsequent CNP-stimulated cGMP accumulation in both C6 and GPNT cells, whereas nitric-oxide-dependent cGMP accumulation was not affected. CNP-stimulated cGMP efflux from C6 glioma cells was significantly reduced under conditions of hyperammonaemia, potentially via a mechanism involving changed in phosphodiesterase expression. Hyperammonaemia-stimulated ROS production was unaffected by CNP but enhanced by a nitric oxide donor in C6 cells. Extracellular vesicle production from C6 cells was enhanced by hyperammonaemia, and these vesicles caused impaired CNP-stimulated cGMP signalling in GPNT cells. Collectively, these data demonstrate functional interaction between CNP signalling and hyperammonaemia in C6 glioma and GPNT cells, but the exact mechanisms remain to be established.

## 1. Introduction

The natriuretic peptides are a highly conserved family of peptide hormones, primarily involved in cardiovascular function [1,2]. Whilst Atrial- and B-type natriuretic peptides (ANP and BNP) are established as regulators of cardiac function and blood pressure, C-type natriuretic peptide (CNP) acts as a local regulator in a range of tissues and is the predominant natriuretic peptide of the central nervous system [2,3]. CNP acts via its selective membrane-bound guanylyl cyclase receptor-B, GC-B/*Npr2*, to increase the formation of cGMP in its target tissues [2]. Additionally, CNP also binds to the clearance receptor/*Npr3*, which was thought to act predominantly to remove circulating natriuretic peptides for intracellular degradation, but in some tissues been shown to signal through G-proteins [4,5]. CNP and GC-B are known to be active in neurons [6], astrocytes and a range of CNS-derived cell lines e.g., C6 glioma cells, LHRH neurons [7], and their activity is subject to regulation by other peptide hormones (e.g., endothelin) [8].

CNP is broadly distributed throughout peripheral and central tissues, where it acts predominantly as an autocrine/paracrine regulator [2,3]. Despite this tissue distribution, understanding of the main biological functions for CNP remains relatively limited, but includes key roles in skeletal growth [9,10,11], meiosis inhibition [12], and more recently in axonal development and bifurcation [13,14]. Whilst Npr2 and PKG signalling are essential for bifurcation of sensory neurons, NO-cGMP signalling is not, suggesting compartmentalized roles for cGMP [15]. In addition, CNP and cGMP are important for learning and memory, by manipulating synaptic plasticity [16]. GC-A and GC-B receptors are expressed during neurodevelopment, with GC-B thought to play a role in neurogenesis [17], but natriuretic peptides also regulate neuronal function in adults. CNP, via the GC-B receptor, has neuroprotective effects in neonatal brain injury models [18]. Furthermore, reduced production of CNP has been observed in patients with neurological disorders, such as Parkinson’s disease [19] and epilepsy [20]. This might reflect CNP-dependent changes in blood-brain-barrier permeability, as shown in bovine brain microvascular endothelial cells and astrocytes [21]. However, understanding of the regulation of CNP and GC-B signalling in cells of the nervous system remains limited.

Astrocytes are the most numerous cell type of the central nervous system, and perform a variety of roles including immune mediation, and are regulators of blood-brain-barrier permeability, in conjunction with endothelial cells and pericytes [22]. A further function of astrocytes is that of ammonia detoxification, whereby ammonia is converted to glutamine and thus protects the brain from ammonia toxicity [23]. This critical process is affected in humans with various neurological conditions, including ageing and Alzheimer’s disease [24], where hyperammonaemia (elevated ammonia in the blood) can be observed. Additionally, humans and companion animals present with liver abnormalities, such as portosystemic shunts, or liver failure (through toxicity or infection), and these can lead to the neurological condition of hepatic encephalopathy (HE) [23,24]. This life-threatening condition can present as ataxia, mood alterations, seizures (e.g., status epilepticus), neuroinflammation, coma and eventually death [25,26,27]. A variety of pharmacological therapies are used to treat HE and these are typically thought to be safe and beneficial in controlling signs [28,29]. However, the evidence for the benefit of treatments targeting ammonia in improving signs of HE in people with liver cirrhosis is weak and reduction in ammonia concentration is not always associated with significant improvements in HE [30]. Furthermore, some drug therapies to treat HE are, themselves, hepatotoxic, which further complicates their use in treating a patient with abnormal liver function, or liver failure [31]. Several of the clinical signs of HE are associated with a reduction in cGMP levels (e.g., cognitive dysfunction, oedema) [27,32,33,34], which has led to the use of cGMP-elevating agents as potential therapeutic targets [35,36]. Disrupted cGMP signalling is also a feature of Alzheimer’s disease, major depressive disorders, multiple sclerosis and Huntington’s disease [37,38,39,40]. Multiple different phosphodiesterases (PDEs) are implicated in these disorders, including PDE4, PDE5 and PDE11, which has driven interest in PDE inhibitors as therapeutics. However, due to the different subtypes and splice variants, with different selectivities and sensitivities for cAMP and cGMP, there are at least 100 different PDE proteins that could potentially be involved in fine-tuning these ubiquitous signalling pathways. Therefore, understanding the complex cyclic nucleotide pathways affected by these disorders is necessary to improve future treatment options.

In contrast to ANP and BNP, CNP is only mildly natriuretic, exerting minimal effects on blood pressure at the local, rather than systemic, level [5]. Furthermore, various long-acting analogues of CNP have progressed through clinical trials, for the treatment of rare disorders of endochondral ossification (such as achondroplasia), where it is a promising treatment for short stature [10,41,42], and CNP can act as an anti-inflammatory agent in endothelial cells [43], hepatic fibrosis [44], and is an inflammatory biomarker in Parkinson’s disease [45]. Here, we examine the effects of conditions mimicking hyperammonaemia on CNP signalling in the well-characterised rat C6 glioma cell line, and the rat brain endothelial GPNT cell line to establish how the natriuretic peptide system in these cell types could be affected by exposure to neurotoxins and in patients with various neurological disorders.

## 2. Materials and Methods

### 2.1. Materials

Ammonium chloride, atrial natriuretic peptide-28 (referred to as ANP), CNP-22 (referred to as CNP), sodium nitroprusside (SNP) and all other chemicals were purchased from Sigma (Sigma-Aldrich, Poole, UK) unless otherwise stated. IL1β, IL6, TNFα and CRP were purchased from R&D Systems (Abingdon, Oxfordshire, UK), and were used as 10 ng/mL, as described previously [46,47,48]. Ammonium chloride was used at concentrations between 1 and 10 mM, as described previously [49,50,51,52,53].

### 2.2. Cell Culture

Rat C6 glioma cells, kindly provided by Dr C Thomas (Cancer Research UK, London, UK), were grown in monolayer culture in DMEM supplemented with high glucose (4500 mg/L) containing 10% (*v/v*) FCS, 1% (*v/v*) antimycotic/antimicrobial, as previously described [54]. Cells were passaged twice weekly and incubated at 37 °C in a humidified 5% (*v/v*) CO_2_/95% (*v/v*) air incubator. For experiments, C6 cells were plated at a density of 1.5 × 10^5^ cells/well in 12-well plates (for RNA extraction and cGMP assays), or 2 × 10^3^ cells/well in 96-well plates (for cell proliferation assays and ROS assays). C6 cells were used from a range of passages, between 8 and 45. In some instances, images of treated C6 cells were taken using a Zeiss Axiovert 135 inverted light microscope. GPNT cells were grown in monolayer culture (as described previously [55], and kindly provided by Prof. John Greenwood (UCL, London, UK), and cultured in Ham’s F10 with Glutamax 10% (*v/v*) FCS, 2 ng/mL bFGF, 80 µg/mL heparin, 1% (*v/v*) antimycotic/antimicrobial. Cells were plated onto collagen I-coated plasticware, and used at a range of passages, between 7 and 20.

### 2.3. Crystal Violet Cell Proliferation Assays

After plating, cells were left to adhere overnight, before stimulation with a range of concentrations (0 to 10 mM) of the ammonia donor, NH_4_Cl, in the absence or presence of 100 nM CNP, in DMEM supplemented with 1% (*v*/*v*) FCS and 1% (*v/v*) antimycotic/antimicrobial, to reduce the basal rate of proliferation. At the indicated time points (24, 48 and 72 h), cells were washed twice with PBS, before being fixed with 4% (*w*/*v*) paraformaldehyde-containing PBS and stored at 4 °C prior to staining with 0.075% (*w*/*v*) crystal violet solution for 15 min, followed by subsequent washing in water and drying overnight. The stained cells were dissolved in 10% (*v*/*v*) acetic acid and left for 30 min, before measuring the optical density at 595 nm using a Mithras LB940 Multimode Plate reader (Berthold, Harpenden, UK). Experiments using GPNT cells also included treatments in the absence and presence of 1 mM SNP.

### 2.4. Cyclic GMP Enzyme Immunoassay

C6 and GPNT cells were allowed to adhere overnight, before being stimulated in physiological saline solution (PSS; 127 mM NaCl, 1.8 mM CaCl_2_, 5 mM KCl, 2 mM MgCl_2_, 0.5 nM NaH_2_PO_4_, 5 mM NaHCO_3_, 10 mM glucose, 0.1% (*w*/*v*) BSA, 10 mM HEPES, adjusted to pH 7.4) with the relevant treatments. All pre-treatment experiments were performed in the absence of 3-isobutyl-1-methylxanthine (IBMX), whereas stimulations were performed in the presence of 1 mM IBMX. Reactions were terminated by addition of ice-cold 100% (*v*/*v*) ethanol, and samples were dried down under vacuum as described previously [56]. For spent media experiments, cells were stimulated with 1% (*v*/*v*) FCS-containing DMEM and the relevant treatments in the absence of IBMX, and spent media were measured without extraction. Reagents and standards were prepared by using instructions within the cGMP enzyme-immunoassay kit (R&D Systems, Abingdon, UK). The optical density of each sample at 450 nm was determined using a Berthold Technologies Mithras LB940 plate reader with MicroWin 4.40 associated software (Berthold Technologies, Harpenden, UK). In some cases, data were normalised to facilitate pooling of multiple experiments (for presentation as either % or control, or fold increases. The range of cGMP concentrations of these normalised responses ranged from 74 to 361 pmol/mL (for CNP), 240 to 725 pmol/mL (for SNP), and 90 to 395 pmol/mL (for 24 h), 198 to 835 pmol/mL (for 48 h) and 368 to 1270 pmol/mL (for 72 h) in the spent media experiments; for GPNT cGMP experiments, the range of cGMP concentrations for these normalised responses was 123 to 494 pmol/mL.

### 2.5. Tissue Collection, RNA Extraction and Multiplex GeXP RT-qPCR Assay

Whole brain tissue was collected from 3 male Sprague Dawley rats, placed into foil and snap frozen in liquid nitrogen. Total RNA was extracted from brain tissue or cultured rat C6 cells using RNAbee reagent (AMS Biotechnology, Abingdon, Oxford, UK), and subjected to DNase treatment (Qiagen, Poole, UK), as described previously [57]. RNA concentrations were determined using ND-100 spectrophotometer (Nanodrop, Thermo Fisher, Hemel Hempstead, UK). Customised GeXP multiplex assays were designed, to detect astrocyte and natriuretic peptide gene targets (*Npr1*, *Npr2*, *Npr3*, *Gfap, S100b, Hmox1, Gad1, Gad2, Fos*), or cGMP target genes (*Pde1a, Pde4a, Pde4b, Pde4d, Pde5a, Pde7a, Pde8a, Pde9a, Pde10a, Pde11a, Mrp4, Mrp5, Prkg1, Prkg2*), with *Actb* used as a housekeeping gene for normalisation (Appendix A). In all assays, 100 ng of total RNA was used per sample. Target-specific reverse transcription and PCR amplification was performed as previously described [57] and in accordance with manufacturer’s instructions (Beckman Coulter, High Wycombe, UK). Briefly, a master mix was prepared for reverse transcription reactions as detailed in the GeXP Starter Kit (AB Sciex, Warrington, Cheshire, UK), and performed using a G-Storm GS1 thermal cycler, using the programme protocol: 48 °C for 1 min, 42 °C for 60 min, and 95 °C for 5 min. From this, an aliquot of each reverse transcription reaction was added to PCR master mix containing GenomeLab kit PCR master mix (AB Sciex, Warrington, Cheshire, UK), and Thermoscientific Thermo-Start Taq DNA polymerase (Thermo Fisher; AB Sciex, Warrington, Cheshire, UK). PCR reaction was performed using a 95 °C activation step for 10 min, followed by 35 cycles of 94 °C for 30 s, 55 °C for 30 s and 70 °C for 60 s. Products were separated and quantified using the GeXP CEQ™ 8000 Genetic Analysis System AB Sciex, Warrington, Cheshire, UK), and GenomeLab Fragment Analysis software (eXpress Analysis Version 1.0.25, Beckman Coulter, UK, Ltd. High Wycombe, Buckinghamshire, UK).

### 2.6. Reactive Oxygen Species (ROS) Assays

C6 cells were cultured on two white-bottomed 96-well plates (2000 cells/well) and left overnight to adhere. ROS assays were performed as described previously [58]. DMEM was discarded and dihydrorhodamine-1,2,3 (DHR-1,2,3) added to each well for 30 min, prior to stimulation with 0.10 nM NH_4_Cl, 100 nM CNP, 1 mM SNP, NH_4_Cl+CNP or NH_4_CL+SNP. All plates were incubated at 37 °C in the dark, prior to absorbance spectroscopy (using a Wallac 1420 plate-reader, with excitation/emission at 485/520 nm) at various timepoints. In some cases, data were normalised to facilitate pooling of multiple experiments (for presentation as fold increases). The range of ROS values for these normalised responses varied from 119,860 to 195,038 ALU (for CNP), and from 23,598 to 29,423 ALU (for SNP).

### 2.7. Extracellular Vesicle Preparation and Flow Cytometry

Spent media from 0 or 10 mM NH_4_Cl-treated C6 cells were briefly centrifuged for 5 min at 4 °C and 3000× *g* to remove cell debris, before storage at −80 °C to await analysis. These supernatants were thawed on ice, and then centrifuged at 4 °C and 17,000× *g* for 15 min, before being resuspended in either serum free media (for stimulating GPNT cells), or Annexin V buffer (Thermo Fisher Scientific, Dartford, UK) for FACS flow cytometric analysis. EVs were stained with or without PE Cy7 Annexin V (to detect exposed phosphatidyl serine), from eBioscience (Thermo Fisher Scientific, Dartford, UK), as described previously [58]. Samples were acquired for 2 min on a FACS Canto II (BD Biosciences; Wokingham, Berkshire, UK), and EVs were counted by reference to enumeration beads (Flow count beads; Beckman Coulter, High Wycombe, UK). Data were normalised to facilitate pooling of multiple experiments (for presentation as fold increases); the range of total EV values for these normalised recordings varied from 549 to 2228 events, and whereas Cy7 annexin EVs ranged from 77 to 278 events.

### 2.8. Data Presentation and Statistical Analysis

Data are shown as representative or pooled from multiple experiments and normalized (as indicated, and typically presented as means ± SEM, or medians (with 5 to 95% confidence intervals). Numerical data were subjected to ANOVA or Mann-Whitney (for nonparametric data), followed by Tukey’s or Dunnett’s multiple comparison tests (where appropriate), accepting *p* < 0.05, using in-built equations in GraphPad Prism 7.0a for Mac (GraphPad, San Diego, CA, USA).

## 3. Results

### 3.1. Molecular and Functional Characterisation of Natriuretic Peptides in Rat C6 Glioma Cells

Previous studies have shown rat C6 glioma cells to respond to natriuretic peptides, as determined by cGMP production [59,60]. To confirm the expression and function of the natriuretic peptide system in C6 cells, we used multiplex RT-qPCR to examine the expression of *Npr1*, *Npr2* and the glial cell marker, *Gfap*, in rat brain tissue and C6 glioma cells. As shown (Figure 1A), rat brain expressed transcripts for all three genes, but C6 cells failed to express detectable *Npr1*. We confirmed this natriuretic peptide receptor expression profile at the functional level by stimulating C6 cells with physiological saline solution (PSS) containing either 0 or 100 nM ANP or CNP for 1 h, in the presence of 1 mM IBMX. As shown (Figure 1B), C6 cells failed to respond to ANP, but CNP caused a significant increase in cGMP accumulation (from 10.8 ± 2.1 pmol/mL to 36.0 ± 3.6 pmol/mL, **** *p* < 0.0001). In subsequent experiments, CNP showed a concentration-dependent stimulation of cGMP accumulation in C6 cells (Figure 1C; from a basal value of 2.5 ± 0.7 pmol/mL to 78.5 ± 10.7 pmol/mL (*** *p* < 0.001) and 133.5 ± 8.3 pmol/mL (**** *p* < 0.0001) for 10 nM and 100 nM CNP, respectively; EC_50_ ~ 7 nM).

Previous studies suggest that hyperammonaemia can reduce cGMP in rat cerebral cortices [33]. Therefore, we next examined if conditions associated with neuroinflammation, such as hyperammonaemia and elevated inflammatory cytokines could affect CNP-stimulated cGMP accumulation in C6 cells. Cells were pre-treated with either 0 (control), 10 mM NH_4_Cl (HA), or with 10 ng/mL of either IL1β, IL6, TNFα or CRP for 24 h prior to subsequent stimulation with 0 or 100 nM CNP for 15 min in the presence of 1 mM IBMX. As shown (Figure 1D), only conditions mimicking hyperammonaemia significantly reduced CNP-stimulated cGMP accumulation (to 46.0 ± 8.7% of control, *p* < 0.05). To determine whether this inhibitory effect of hyperammonaemia was concentration-dependent, C6 cells were pre-treated with either 0, 1, 5 or 10 mM NH_4_Cl (HA) for 24 h prior to subsequent stimulation with 0 or 100 nM CNP for 15 min in the presence of 1 mM IBMX. The effect of hyperammonaemia was shown to be concentration dependent, as C6 cells pre-treated with 5 mM and 10 mM NH_4_Cl for 24 h showed a significant reduction in cGMP accumulation (to 63.2 ± 16.5% (* *p* < 0.05) and 62.0 ± 6.4% (** *p* < 0.01) of control response to CNP, for 5 mM and 10 mM NH_4_Cl, respectively) following a subsequent 15 min stimulation with 100 nM CNP (Figure 1E).

### 3.2. Effects of Hyperammonaemia and CNP on Cell Proliferation in Rat C6 Cells

Having established that C6 cells were functionally responsive to CNP and sensitive to conditions of hyperammonaemia, the next experiments examined the effects of exposing C6 cells to increasing concentrations of NH_4_Cl on cell proliferation. C6 cells were plated at 2 × 10^3^ cells/well and left to adhere overnight before transferring to media containing 1% (*v*/*v*) FCS and the indicated concentration of NH_4_Cl (ranging from 0 to 10 mM), in the absence or presence of 100 nM CNP. Cells were fixed at the indicated time points and cell proliferation was determined by crystal violet assay. As shown (Figure 2A,B,D), cell proliferation was not significantly affected by conditions of hyperammonaemia or the presence of CNP, after 24 h and 48 h. However, by 72 h (Figure 2C), C6 cells exposed to 10 mM NH_4_Cl showed significantly reduced cell proliferation (from an optical density of 0.62 ± 0.04 to 0.43 ± 0.03, * *p* < 0.05), an effect that was not altered in the presence of 100 nM CNP (from an optical density of 0.63 ± 0.01 to 0.34 ± 0.05, ** *p* < 0.01). Cell morphology was assessed by light microscopy after the 72 h treatments, and confirmed cell density in the presence of hyperammonaemia, and pronounced thinning of processes and rounding up of cells (Figure 2E), effects that were not prevented by the presence of CNP.

### 3.3. Effects of Hyperammonaemia on CNP-Stimulated and Sodium Nitroprusside-Stimulated cGMP Accumulation in Rat C6 Cells

Having established that hyperammonaemia, but not inflammatory cytokines, can inhibit CNP-stimulated cGMP accumulation, we next examined the kinetics of this response. C6 cells were plated at 1.5 × 10^5^ cells/well and left to adhere overnight before transferring to media containing 1% (*v*/*v*) FCS and either 0 or 10 mM NH_4_Cl, to create conditions of hyperammonaemia, for either 1 h or 24 h. Subsequently, the media were replaced with PSS containing 1 mM of the non-selective phosphodiesterase inhibitor, IBMX, in the absence or presence of 100 nM CNP for up to 15 min, prior to termination of the reaction and assay for cGMP accumulation. As shown (Figure 3A), CNP caused a rapid increase in total cGMP accumulation within 5 min, an effect that was continued by 15 min. However, pre-treatment in hyperammonaemic conditions caused a significant inhibition to CNP-stimulated cGMP accumulation, at both pre-treatment time points 1 h (to 65.2 ± 11.0% of control response, ** *p* < 0.01) and 24 h (to 66.2 ± 5.3% of control response, * *p* < 0.05)). In addition, the rate of cGMP accumulation appeared to reduce after 5 min of the CNP stimulation (Figure 3C), with only the control response to CNP maintaining significant linear accumulation (r^2^ = 0.64***).

Having shown that cGMP accumulation, mediated via CNP-stimulation of the GC-B receptor was sensitive to conditions of hyperammonaemia, similar experiments were performed on C6 cells pre-treated with the identical pre-treatment paradigm, but subsequently stimulated with the nitric oxide donor, Sodium Nitroprusside (SNP, 1 mM) for up to 15 min. As shown (Figure 4A,B), SNP caused a rapid and sustained increase in cGMP accumulation in C6 cells. Furthermore, pre-treatment with 10 mM NH_4_Cl failed to significantly alter the amount of SNP-stimulated cGMP accumulation. In contrast to the CNP responses (Figure 3C), the rate of cGMP accumulation was significantly maintained between 5 and 15 min of the SNP stimulation (Figure 4C), with only the HA (24 h) response to CNP failing to remain linear (r^2^ = 0.63).

### 3.4. Effects of Hyperammonaemia on CNP-Stimulated cGMP Efflux from Rat C6 Cells

A reduction in cGMP signalling is associated with several clinical signs linked to Alzheimer’s disease [61], major depressive disorders [40], and hepatic encephalopathy [27]. To determine whether conditions of hyperammonaemia could cause a similar inhibition of extracellular cGMP levels released from C6 cells, we examined the effect of NH_4_Cl pre-treatment on cGMP efflux between 24 and 72 h. C6 cells were plated at 1.5 × 10^5^ cells/well, and left to adhere overnight before transferring to media containing 1% (*v*/*v*) FCS and either 0 or 10 mM NH_4_Cl, in the absence or presence of 100 nM CNP, for up to 72 h. At each time point, spent media were removed and stored at −80 °C, prior to assay for extracellular cGMP (as an indication of cGMP efflux). As shown (Figure 5), extracellular cGMP concentrations remained unaffected in cells treated in the absence of CNP; however, CNP caused significantly enhanced cGMP accumulation at each time point, an effect that was inhibited in the presence of hyperammonaemia (from 9.4 ± 1.8-fold to 3.5 ± 0.7-fold (** *p* < 0.0001), and from 12.5 ± 2.6-fold to 1.9 ± 0.7-fold (** *p* < 0.0001), after 48 h and 72 h, respectively).

### 3.5. Effects of Hyperammonaemia on Expression of Genes Associated with cGMP Regulation in Rat C6 Cells

As the inhibitory effects of hyperammonaemia on cGMP accumulation and efflux might reflect alterations to the expression of multiple genes associated with the production and metabolism of cGMP, we examined how the expression of 15 cGMP-associated transcripts were affected. Total RNA was harvested from the same C6 cells that had been treated with 0 or 10 mM NH_4_Cl for 48 h. Custom-designed multiplex RT-qPCR assays were initially performed to examine the expression of cyclic nucleotide phosphodiesterases (PDEs), multidrug resistance proteins (MRPs), protein kinase G (PKG) and natriuretic peptide receptors (specifically Pde1a, Pde4a, Pde4b, Pde4d, Pde5a, Pde7a, Pde8a, Pde9a, Pde10a, Pde11a, Mrp4, Mrp5, Prkg1, Npr2 and Npr3). As shown (Figure 6A,B), conditions of hyperammonaemia caused a significant increase in the expression of Pde5a, Pde11a, Mrp4 and Npr3 compared with control cells (from 1.0 ± 0.05 to 1.3 ± 0.1 (* *p* < 0.05), 1.0 ± 0.1 to 1.5 ± 0.06 (** *p* < 0.01), 1.0 ± 0.03 to 1.4 ± 0.14 (* *p* < 0.05), and 1.0 ± 0.09 to 1.21 ± 0.1 (* *p* < 0.05), for Pde5a, Pde11a, Mrp4 and Npr3, respectively). Using an additional multiplex RT-qPCR assay, we also examined changes in expression of astrocyte enriched transcripts (Gfap, S100, Gad1, Gad2) as well as an inflammatory response gene (Hmox1), and an immediate early gene (cFos) in these same RNA samples. As shown (Figure 6B), conditions of hyperammonaemia caused a significant increase in the expression Hmox1, Gad1 and Fos compared with control cells (from 1.0 ± 0.06 to 1.6 ± 0.13 (*** *p* < 0.001), 1.0 ± 0.04 to 1.6 ± 0.08 (**** *p* < 0.0001), and 1.0 ± 0.09 to 2.2 ± 0.24 (** *p* < 0.01), for Hmox1, Gad1 and cFos, respectively).

### 3.6. Effects of Hyperammonaemia on Reactive Oxygen Species (ROS) Production in Rat C6 Cells

In several systems, CNP acts as an anti-inflammatory agent [43,44,62]. Part of the progression of inflammation is the production of reactive oxygen species, and markers of oxidative stress have been observed in patients with hepatic encephalopathy [63]. In the next series of experiments, we examined the potential effects of hyperammonaemia, CNP and SNP on ROS production in C6 cells. Cells were plated at 2 × 10^3^ cells/well and left to adhere overnight before switching to 1% (*v*/*v*) FCS-containing media in the presence of Dihydrorhodamine-1,2,3 (DHR) for 30 min, before being treated with 0 (control), 10 mM NH_4_Cl (HA), 100 nM CNP or both HA and CNP. As shown (Figure 7A,B), ROS production increased over time, and was modestly, but significantly increased under conditions of mimicking hyperammonaemia (to 1.18 ± 0.02-fold (**** *p* < 0.0001) and 1.21 ± 0.02-fold (**** *p* < 0.0001), for HA and HA+CNP, respectively), an effect that was not altered in the presence of CNP. However, when similar experiments were performed, using 1 mM SNP instead of CNP (Figure 7C,D), the presence of the nitric oxide donor alone caused a significant increase in ROS production that was much greater than that caused by conditions mimicking hyperammonaemia (to 2.4 ± 0.09-fold (**** *p* < 0.0001) and 2.80 ± 0.1-fold (**** *p* < 0.0001) for SNP and HA+SNP, respectively). Collectively, these finding suggest that activation of the NO/cGMP pathway by SNP in C6 cells can cause significant ROS production, even in the absence of conditions of hyperammonaemia.

### 3.7. Effects of Hyperammonaemia on the Natriuretic Peptide System in Rat GPNT Brain Endothelial Cells

The blood brain barrier consists of astrocytes, pericytes and endothelial cells [64]. In order to establish whether the inhibitory effects of hyperammonaemia on natriuretic peptide signalling observed in C6 glioma cells was also seen in rat brain endothelial cells, we repeated pharmacological and cellular experiments in rat GPNT brain endothelial cells. As shown (Figure 8A), transcripts for both *Npr1* and *Npr2* receptors were detected by multiplex RT-qPCR. This expression profile was confirmed at the pharmacological level, with both ANP and CNP stimulating cGMP accumulation in GPNT cells (Figure 8B), with CNP activating the GC-B receptor in a concentration-dependent manner (Figure 8C; EC_50_ ~ 34 nM). In contrast, 1 mM SNP failed to stimulate cGMP accumulation (Figure 8B).

We next examined how conditions mimicking hyperammonaemia might affect GPNT responsiveness to cGMP signalling. In contrast to the observed inhibitory effects of hyperammonaemia on cell proliferation in C6 cells, GPNT cell proliferation was not affected by treatment with a range of concentrations of NH_4_Cl, either in the absence or presence of 100 nM CNP or 1 mM SNP (Figure 8D–F). Finally, we determined whether conditions of hyperammonaemia could alter CNP-stimulated cGMP accumulation in GPNT cells. As shown (Figure 8G,H), pre-treatment of GPNT cells with 10 mM NH_4_Cl for either 1 h or 24 h significantly inhibited CNP-stimulated cGMP accumulation (to 54.5 ± 11.5% (** *p* < 0.01) after 1 h of hyperammonaemia pre-treatment, and to 58.0 ± 3.7% (**** *p* < 0.001) after 24 h of hyperammonaemic conditions. Collectively, these data show GPNT brain endothelial cells demonstrate common and distinct sensitivities to conditions of hyperammonaemia compared with C6 glioma cells.

### 3.8. Effects of Hyperammonaemia on Extracellular Vesicle Production from C6 Cells and Their Functional Effects on Natriuretic Peptide Signalling in GPNT Cells

Extracellular vesicle production is associated with various metabolic, cardiovascular and endocrine pathologies [58,65], and their production has been demonstrated from astrocytes and C6 cells [66,67]. In addition, rats fed a manipulated diet to induce chronic hyperammonaemia, peripherally produce extracellular vesicles that promote neuroinflammation in astrocytes [68]. Initially, to establish whether extracellular vesicles production from C6 cells was affected by conditions of hyperammonaemia, we stimulated C6 cells in serum-free media containing 0 or 10 mM NH_4_Cl for up to 72 h, before harvesting the spent media, removing cellular debris, and centrifuging to pellet extracellular vesicles. Following staining for exposed phosphatidyl serine with Cy7-labelled Annexin-5, extracellular vesicles were examined by flow cytometry (Figure 8A). As shown (Figure 8B,C), hyperammonaemia failed to alter the number of Annexin 5-positive extracellular vesicles produced by C6 cells, but did significantly increase the total number of extracellular vesicles produced within 24 h (to 124.0 ± 8.0%, * *p* < 0.05).

Finally, the functional effects of C6-derived extracellular vesicles (from control or 10 mM NH_4_Cl-treated cells) were investigated, by using them to pre-treat GPNT cells for 24 h, prior to determining the effect on CNP-stimulated cGMP accumulation. As shown (Figure 9D), 100 nM CNP significantly stimulated cGMP accumulation in GPNT cells treated with either control (0 mM) or hyperammonaemia (10 mM)-derived extracellular vesicles (to 56.7 ± 8.0 and 29.5 ± 7.3 pmol/mL, **** *p* < 0.0001 and ** *p* < 0.01, respectively). However, GPNT cells pretreated with hyperammonaemia-derived extracellular vesicles generated significantly less cGMP in response to CNP (reduced to 52.6 ± 12.8% of control, ** *p* < 0.01). These data suggest that hyperammonaemia-derived extracellular vesicles from C6 cells functionally inhibit CNP-stimulated cGMP signalling in GPNT cells.

## 4. Discussion

The CNP/GC-B pathway is the major natriuretic peptide system in the central nervous system, with CNP having effects on multiple different cell types [2,17], to influence key processes such as neurogenesis, axonal bifurcation, learning and memory [13,14,15,16]. Despite this, understanding of how the CNP/GC-B system in cells of the CNS is affected under pathological conditions remains limited. Excess concentrations of ammonia (hyperammonaemia) creates neurotoxicity, and is a feature of neurological disorders such as Alzheimer’s disease, ageing, and hepatic encephalopathy [23,24,25]. Limited studies suggest that hyperammonaemia can inhibit CNP signalling in rat cerebral cortices [33] and the RBE-4 endothelial cell line [34], but a more detailed investigation of the effects of ammonia on CNP is needed. Several previous studies have shown the importance of cGMP signalling in mediating some of the cognitive aspects of major depressive disorders, multiple sclerosis, Huntington’s disease, as well as hepatic encephalopathy, and suggests that manipulation of cGMP/PDE/PKG pathway represents a plausible therapeutic target [27,32,33,34,35,36,37,38,39,40]. Our current study demonstrates that cGMP generation mediated via membrane guanylyl cyclases (specifically GC-B/*Npr2*) is sensitive to hyperammonaemia, but not inflammatory cytokines, by a mechanism that does not appear to affect cGMP generation through soluble guanylyl cyclase activation.

Here, we have utilized the well-characterised C6 glioma cell line (that has phenotypic features in common with astrocytes), and GPNT cells (as a model of brain endothelial cells) [69]. C6 cells have been widely used as model astrocytes for several decades, and despite the limitations of being a tumour cell line (as opposed to primary culture), repeated studies have shown that key aspects of astrocyte biology are maintained in their gene expression and functional profiles [70,71], making them an appropriate tool for fundamental investigations. The expression data within our current study detected transcripts for astrocyte-enriched transcripts in C6 cells, such as *Gfap*, *S100*, *Gad1* and *Gad2*, albeit with lower abundance than that seen in primary rat brain tissue, which is likely a reflection of the undifferentiated nature of C6 cells compared to primary astrocytes. C6 cells have previously been shown to respond to natriuretic peptides, to stimulate cGMP accumulation, and enhance expression of immediate early genes [59,60]. However, it is important to acknowledge that key functional differences exist between primary astrocytes and C6 glioma cells, including how these two populations of cells metabolise ammonia [72]. Therefore, it remains to be established as to how the natriuretic peptide system is altered by hyperammonaemia in primary astrocytes, as well as in vivo.

The expression and pharmacology data described herein support previous studies that showed that CNP-stimulated cGMP accumulation via the GC-B receptor predominates, rather than ANP activity via the GC-A receptor [59], and our estimated EC_50_ values for CNP activity in C6 cells (~7 nM) are almost identical to those previously published (~10 nM) [8]. Astrocytes interact with endothelial cells to form the blood brain barrier, and as such, make an attractive target cell in which to examine the effects of CNP, since its function as a regulator of the endothelium has been previously demonstrated [2]. CNP is the major natriuretic peptide of the CNS, and the CNP-selective GC-B/*Npr2* receptor is expressed almost ubiquitously throughout the brain and endothelial cells [2,17]. For these reasons, we focused our subsequent investigations on how CNP function might be altered under conditions of hyperammonaemia.

Cyclic GMP signalling in GPNT cells has not been previously reported. Our current study reveals presence of both natriuretic peptide guanylyl cyclase receptors (GC-A/*Npr1* and GC-B/*Npr2*), in contrast to what we observed in C6 glioma cells. However, we failed to detect a significant cGMP response to the nitric oxide donor, SNP, in GPNT cells. The mechanism for this lack of response to SNP remains to be established but may reflect an absence of one of the functional subunits of soluble guanylyl cyclase. The enhanced cGMP response to ANP observed in the current study suggests that GPNT cells are likely to express more GC-A (*Npr1*) receptors than GC-B (*Npr2*). Nevertheless, cGMP accumulation in response to CNP was robust and concentration-dependent, exhibiting an ~EC_50_ value similar to that observed in C6 glioma cells. Therefore, GPNT cells represent a useful additional brain endothelial cell line in which to investigate the roles of natriuretic peptides.

Humans and companion animal patients with neurological disorders are known to have elevated levels of inflammatory markers, bile acids and other toxins, in their circulation [23,24,25]. Of these toxins, one of the most common molecules is ammonia, the concentrations of which typically reach pathological levels and cause conditions of hyperammonaemia. This is of particular concern with regards to astrocyte function, as these cells are the only cell type of the central nervous system capable of metabolizing ammonia, essentially detoxifying the CNS [22]. Hyperammonaemia is a mechanistic component of several of the clinical signs associated with neurodegenerative disorders, ageing and hepatic encephalopathy, including cellular oedema, inflammation, abnormal GABA production, and inhibition of the cGMP pathway. Our current studies initially examined the potential effect of ammonia (in the form of NH_4_Cl) on C6 and GPNT cell proliferation. There was a significant, concentration-dependent inhibition in C6 cell proliferation after 72 h of hyperammonaemic conditions, an effect that was not prevented by the presence of CNP. This inhibitory effect of hyperammonaemia on cell proliferation was not observed in GPNT brain endothelial cells, which may suggest a different sensitivity to ammonia, as has been seen in a comparison of other cell types [73]. Using the crystal violet assay prevents identification of the specific mechanisms underlying this reduction in cell number, such as apoptosis, necrosis, senescence or reduced proliferation. Hyperammonaemia-dependent reductions in cell proliferation and an increase to cellular senescence have previously been reported in primary rat astrocytes [74,75]. In the current study, we did not demonstrate that C6 cells exhibited any form of cell death under conditions of hyperammonaemia, but previous studies have found that 10 mM NH_4_Cl caused delayed apoptosis after 72 h and 96 h [76]. Therefore, it is possible that prolonged exposure to conditions of hyperammonaemia activate apoptosis pathways in C6 glioma cells, but not GPNT cells.

Previous studies have shown that conditions of hyperammonaemia can cause an attenuation to CNP-stimulated cGMP production in rat cerebral cortex sections and RBE-4 cells, although the mechanisms involved are not clear [32,33,34]. Our current findings suggest that CNP-stimulated cGMP accumulation in C6 glioma cells and GPNT brain endothelial cells is subject to inhibition by acute and sustained conditions of hyperammonaemia, as CNP-stimulated cGMP accumulation was reduced by approximately 40%. This degree of inhibition is similar to that reported in the astrocytic compartment of rat cerebral cortex slices [33], and in rat RBE-4 cells [34]. The endogenous GC-B receptors in C6 cells exhibited homologous desensitization in the presence of sustained CNP stimulation, as indicated by the failure to maintain the initial rate of cGMP accumulation, similar to our previous reports in rat GH3 somatolactotropes [77] and murine αT3-1 gonadotropes [78]. Interestingly, this key pharmacological process of tachyphylaxis was also present under conditions of hyperammonaemia. Whether this inhibitory effect of hyperammonaemia represents heterologous desensitization, or an alternative mechanism such as altered receptor binding properties, is unclear. However, as desensitization was also observed in the absence of hyperammonaemia, it is likely that the mechanisms controlling tachyphylaxis and ammonia sensitivity, are independent. When similar experiments were performed but using the nitric oxide donor SNP to stimulate cGMP accumulation, the rate of cGMP accumulation between 5 and 15 min of stimulation with SNP remained significantly linear under almost all conditions. Furthermore, SNP-stimulated cGMP accumulation was insensitive to hyperammonaemia at either 1 h or 24 h exposure. Given that administration of agents that elevate cGMP can alleviate some of the adverse effects of hepatic encephalopathy in in vivo models of the disease [35,36], it is tempting to suggest that activation of the soluble guanylyl cyclase pathway (through NO) is a better therapeutic target than the particulate guanylyl cyclase pathway (through use of analogs of CNP). Nitric oxide production in astrocytes has previously been shown to be stimulated by acetylcholine and glutamine [79], thus providing an alternative mechanism through which endogenous cGMP production could be stimulated. However, NO-stimulated cGMP production in astrocytes has also been shown to lead to cell death in astrocytes [80]. Therefore, any therapeutic use that manipulates cGMP levels in patients with hepatic encephalopathy, or indeed other neurological disorders associated with impaired cGMP production, needs to consider these potentially undesirable effects.

We extended our investigations of cGMP signalling in C6 cells to determine whether hyperammonaemia could alter CNP-stimulated cGMP efflux, as a reduction in extracellular cGMP concentrations is implicated in the cognitive dysfunction in patients with hepatic encephalopathy [27], and reduced plasticity in patients with Alzheimer’s disease and major depressive disorders [37,38,39,40]. Our data show that over a more prolonged time period (48 h and 72 h), conditions of hyperammonaemia significantly reduce CNP-stimulated cGMP efflux from C6 cells. This may partially be explained by the findings of the previous experiments, which indicated that GC-B receptors may either undergo desensitization or down-regulation under conditions of hyperammonaemia. However, our multiplex RT-qPCR assays of cGMP target genes showed an upregulation in transcripts encoding *Pde5a*, and *Pde11a*. PDE5a is a cGMP-binding, cGMP-specific phosphodiesterase [81] whereas as PDE11a acts as a dual-specificity cAMP/cGMP phosphodiesterases [82]; therefore, upregulation of either of these enzymes under conditions of hyperammonaemia would potentially lead to the observed reduction to extracellular cGMP concentrations. In the same multiplex RT-qPCR assays, we also observed a significant increase in the expression of transcripts for *Mrp4* as well as the natriuretic peptide clearance receptor, *Npr3*. An increase in *Npr3* expression is consistent with reduced cGMP levels, as it represents a mechanism by which CNP is diverted from activating GC-B receptors. *Mrp4* encodes the multidrug resistance protein 4 (MRP4) that acts to transport cyclic nucleotides out of the cell [83]. Our observations are similar to those in rat astrocytes and human cerebral cortex from patients with hepatic encephalopathy, where ammonia increased both mRNA and protein expression [84]. Upregulation in MRP4 would be expected to *increase* extracellular cGMP, by virtue of increasing the efflux of cGMP in to the extracellular space. Therefore, our observation of increased *Mpr4* expression under conditions where cGMP concentrations are *decreased*, could represent some form of attempted compensatory mechanism. However, it is important to note that our gene expression data were obtained from C6 cells in the *absence* of CNP-stimulation; therefore, further investigations are required to determine whether hyperammonaemia alters *Pde5a*, *Pde11a* and *Npr3* in the presence of CNP signalling.

In addition to examining changes to genes within the natriuretic peptide/cGMP pathway, we used the same RNA samples to determine the effects of hyperammonaemia on the expression of astrocyte-enriched transcripts (*Gfap, S100, Gad1, Gad2*), a marker of cellular inflammation (*Hmox*) and an immediate early gene (*Fos*), in C6 cells. We observed robust upregulation of *Gad1*, *Fos* and *Hmox* in C6 cells treated with conditions mimicking hyperammonaemia for 48 h but failed to detect any significant alteration in the expression of *Gfap*, *S100* or *Gad2*. Previous studies have reported an inhibitory effect of ammonia on *Gfap* [85,86], although these studies used a different time course to the current study. Similarly, S100 protein secretion from cultured astrocytes was enhanced in the presence of NH_4_Cl, but within 24 h of treatment [87]. Therefore, it is possible that the lack of significant changes to *Gfap* and *S100* expression in the current study reflect a time-dependent sensitivity to hyperammonaemia. Our findings that show an increase in *Gad1* expression in C6 cells treated with NH_4_Cl are in keeping with recent reports of ammonia-induced hyperGABAergic states in zebrafish [88]. Although the gene expression data reported in the current study reveal a significant increase in multiple transcripts, we have not examined the potential mechanisms by which ammonia regulates gene transcription in C6 glioma cells. Previous studies have revealed that conditions of hyperammonaemia can increase the production of inflammatory cytokines (TNFα, IL1β, IL6) through pathways that are sensitive to NF-κB, ERK and ROS signalling [89]. The potential roles for these pathways in mediating the observed changes in gene expression in C6 glioma cells remains to be established.

The increase in *Hmox* expression in C6 cells, following hyperammonaemia treatment for 48 h, is indicative of the inflammatory component of hyperammonaemia (as HMOX acts as an anti-inflammatory agent). This upregulation in *Hmox* expression could be a protective response to the increase in ROS production observed over a 24 h period, in the presence of hyperammonaemia. Co-treatment with CNP was unable to attenuate this increase in ROS, suggesting that CNP does not protect against ROS formation in C6 cells (potentially due to the inhibitory effect that hyperammonaemia has on CNP-stimulated cGMP production). However, despite the fact that SNP-stimulated cGMP production was unaffected by conditions of hyperammonaemia, SNP co-treatment exacerbated hyperammonaemia effects on ROS production, leading to a significantly enhanced inflammatory response. Therefore, the use of nitric oxide promoting agents as potential anti-inflammatory therapies in conditions of hyperammonaemia may not be appropriate.

As well as their contributions to systemic pathologies [58,65], extracellular vesicles may represent pivotal regulators of the neuroimmune response [90]. Our findings confirm the production of extracellular vesicles from C6 cells and astrocytes [66,67], and represent the first demonstration that hyperammonaemia can increase extracellular vesicle production from C6 cells. This complements in vivo studies, using a dietary manipulation approach to induce chronic hyperammonaemia, that showed peripheral production of extracellular vesicles that promote neuroinflammation in astrocytes [68]. Our functional data demonstrate that hyperammonaemia-induced extracellular vesicles from C6 cells are inhibitory to CNP-stimulated cGMP accumulation in GPNT cells. Proteomic analysis of these vesicles is required to establish the identity of possible antagonists of CNP signalling. Nevertheless, the current observations enhance the understanding of cell signalling relationships under conditions of hyperammonaemia.

Reduced cGMP production in the central nervous system is implicated in cellular oedema, cognitive dysfunction, impaired axonal bifurcation and learning deficits [13,14,15,16,27]. Therefore, agents that increase cGMP concentrations remain an attractive approach to treat these clinical signs. Our cellular studies suggest that, despite being a potent activator of cGMP in C6 glioma cells, the use of CNP as one such cGMP-elevating therapy may well be compromised by the inhibitory effects that hyperammonemia has on GC-B function. Our findings that hyperammonemia can also inhibit CNP signalling in rat brain endothelial GPNT cells suggests that cGMP production in multiple components of the central nervous system could be impaired by hyperammonaemia. Therefore, the restoration of cGMP signalling remains an intriguing therapeutic target for multiple neurological conditions.

## Figures and Tables

**Figure 1 cells-10-00398-f001:**
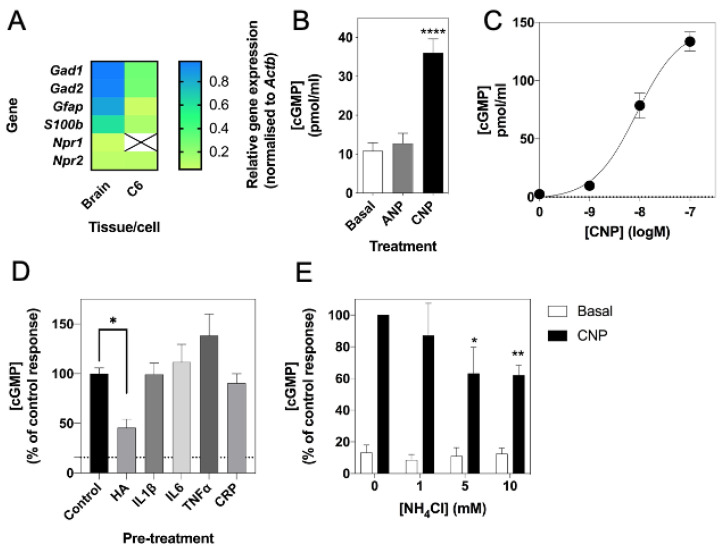
Molecular and pharmacological characterization of natriuretic peptide system in rat C6 glioma cells. (**A**) Multiplex RT-qPCR was performed on RNA extracted from rat brain tissue or C6 cells. The data shown are mean relative gene expression, normalized to *Actb*, of 3 to 6 individual RNA extractions. (**B**) Total cGMP accumulation in C6 cells treated with 0 or 100 nM ANP or CNP for 1 h in physiological saline solution containing 1 mM IBMX, before measuring with a commercially available cGMP-EIA kit (R&D Systems) as described previously. Data shown are means ± SEM pooled from three independent experiments, each performed in triplicate. **** *p* < 0.0001, significantly different from basal. (**C**) Concentration-dependent effects of CNP on cGMP accumulation in C6 cells. Cells were treated with the indicated concentrations of CNP in PSS containing 1 mM IBMX for 1 h. Data shown are means ± SEM, representative of three independent experiments, each performed in triplicate. (**D**) C6 cells were pre-treated in media containing 1% (*v*/*v*) FCS and 0.10 mM NH_4_Cl (HA), or 10 ng/mL of either IL1β, IL6, TNFα or CRP for 24 h prior to subsequent stimulation with 100 nM CNP in the presence of 1 mM IBMX for 15 min. The data shown are means ± SEM of five to twelve individual stimulations, expressed as the percentage of the control response to CNP * *p* < 0.05, significantly different from control response to CNP. The dotted line indicates basal cGMP concentration. (**E**) C6 cells were pre-treated in media containing 1% (*v*/*v*) FCS and 0, 1, 5 or 10 mM NH_4_Cl for 24 h prior to subsequent stimulation with 100 nM CNP in the presence of 1 mM IBMX for 15 min. The data shown are means ± SEM of four to ten independent experiments, each performed in triplicate and are expressed as the percentage of the control response to CNP. * *p* < 0.05, ** *p* < 0.01, significantly different from control response to CNP.

**Figure 2 cells-10-00398-f002:**
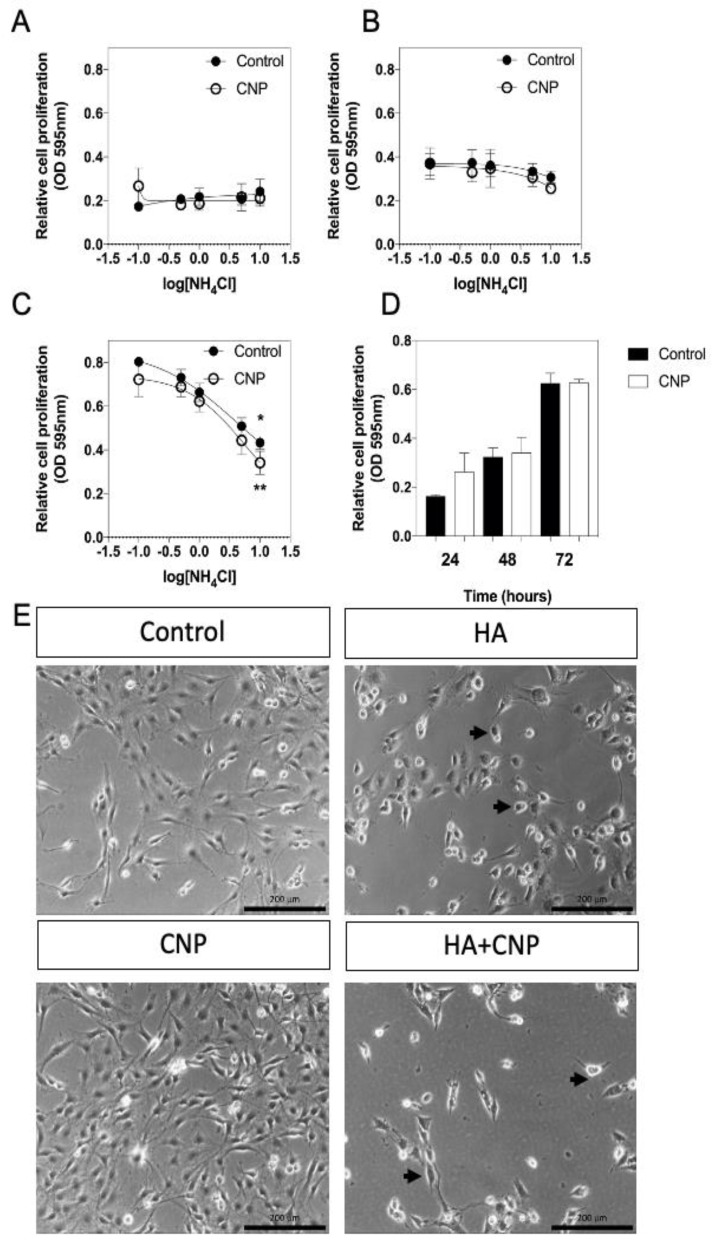
Effects of hyperammonaemia and CNP on cell proliferation and morphology in rat C6 glioma cells. C6 cells were treated with the indicated concentrations of NH_4_Cl (log mM) (as an NH3 donor) in the absence or presence of 100 nM CNP for (**A**) 24 h, (**B**) 48 h or (**C**) 72 h, before being fixed in paraformaldehyde, and stained for crystal violet assay. The data shown are means ± SEM pooled from 3 independent experiments, each performed with 8 replicates. * *p* < 0.05, ** *p* < 0.01, significantly different from untreated cells. (**D**) Cell viability in C6 cells in the presence of 0 or 100 nM CNP over 72 h. The data shown are means ± SEM pooled from 3 independent experiments, each performed with 8 replicates. (**E**) Representative microscopy images of C6 cells treated for 72 h with 0 (control) or 10 mM NH_4_Cl (HA), in the presence of 100 nM CNP alone (CNP) or in combination (HA+CNP). Arrow heads indicated rounded cells.

**Figure 3 cells-10-00398-f003:**
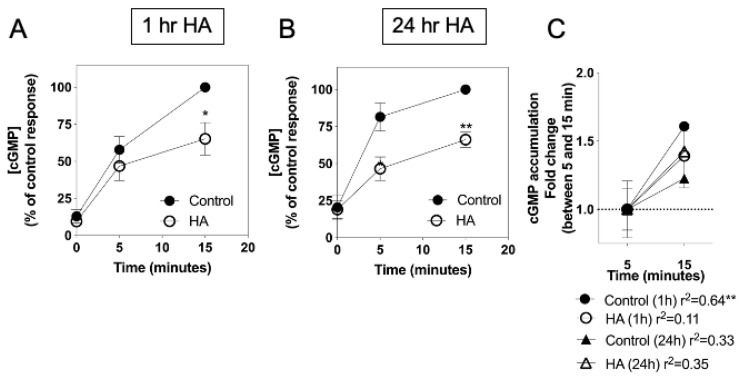
Effects of hyperammonaemia on CNP-stimulated cGMP accumulation in rat C6 glioma cells. Cells were initially treated in media containing 1% (*v*/*v*) FCS and 0 or 10 mM NH_4_Cl (HA) for either (**A**) 1 h or (**B**) 24 h, prior to subsequent stimulation with 100 nM CNP in the presence of 1 mM IBMX for up to 15 min. The data shown are means ± SEM of five to seven independent experiments, each performed in triplicate and are expressed as the percentage of the control; * *p* < 0.05, ** *p* < 0.01, significantly different from control response to CNP. (**C**) Linear accumulation of cGMP between 5 and 15 min of CNP stimulation. Data shown are normalized to cGMP accumulation after 5 min, expressed as means ± SEM of five to seven independent experiments. ** *p* < 0.01, significantly deviates from zero.

**Figure 4 cells-10-00398-f004:**
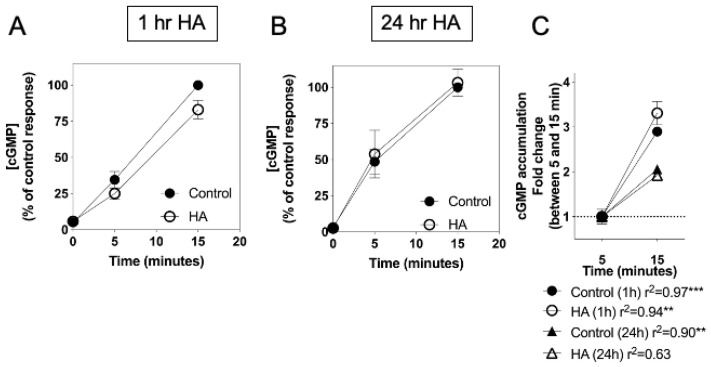
Effects of hyperammonaemia on SNP-stimulated cGMP accumulation in rat C6 glioma cells. Cells were initially treated in media containing 1% (*v*/*v*) FCS and 0 or 10 mM NH_4_Cl (HA) for either (**A**) 1 h or (**B**) 24 h, prior to subsequent stimulation with 1 mM SNP in the presence of 1 mM IBMX for up to 15 min. The data shown are means ± SEM of three independent experiments, each performed in triplicate and are expressed as the percentage of the control response to SNP. (**C**) Linear accumulation of cGMP between 5 and 15 min of SNP stimulation. Data shown are normalized to cGMP accumulation after 5 min, expressed as means ± SEM of three independent experiments. *** *p* < 0.001, ** *p* < 0.01, significantly deviates from zero.

**Figure 5 cells-10-00398-f005:**
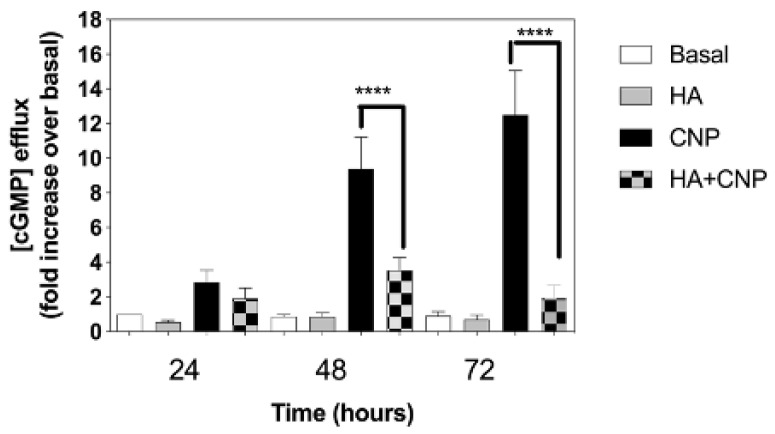
Effects of hyperammonaemia on CNP-stimulated cGMP efflux rat C6 glioma cells. C6 cells were treated in media containing 1% (*v*/*v*) FCS and either 0 or 10 mM NH_4_Cl, (HA) in the absence or presence of 100 nM CNP, for up to 72 h. Spent media were collected at the indicated time points and analysed for cGMP content. Data shown are means ± SEM of 6 to 8 independent experiments, each performed in triplicate, and normalized to basal at each time point. **** *p* < 0.0001, significantly different to control response to CNP.

**Figure 6 cells-10-00398-f006:**
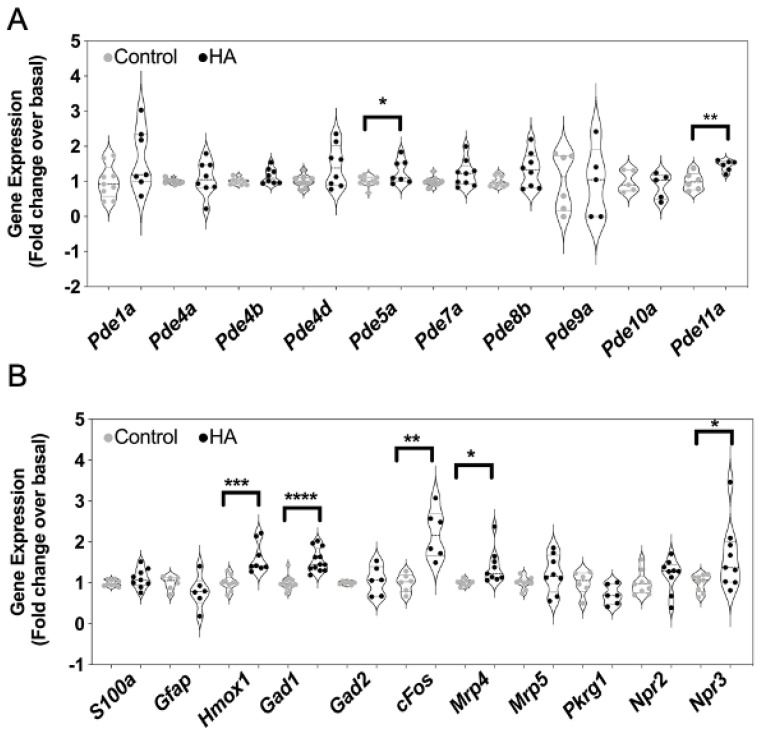
Effects of hyperammonaemia on cGMP-associated, and astrocyte-enriched gene expression in rat C6 glioma cells. Cells were initially treated in media containing 1% (*v*/*v*) FCS and 0 or 10 mM NH_4_Cl (HA) for 48 h, prior to extracting total RNA. Multiplex RT-qPCR assays were performed on RNA extracted from C6 cells to detect (**A**) transcripts encoding phosphodiesterase enzymes, (**B**) other cGMP-associated genes and astrocyte-enriched transcripts. The data shown are the medians (with 5 to 95% confidence intervals) of relative gene expression, normalized to *Actb* and expressed as fold change over basal, from 4 to 12 individual RNA extractions. * *p* < 0.05, ** *p* < 0.01, *** *p* < 0.001, **** *p* < 0.0001, significantly different from control.

**Figure 7 cells-10-00398-f007:**
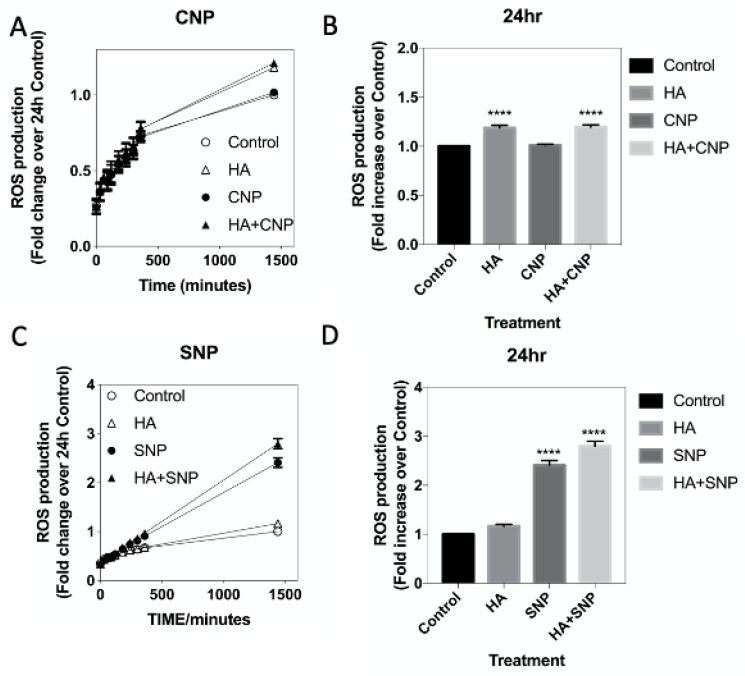
(**A**–**D**). Effects of hyperammonaemia on ROS production in rat C6 glioma cells. Cells were pre-treated in media containing 1% (*v*/*v*) FCS and 100 µM DHR for 30 min prior to stimulation with either 0, 10 mM NH_4_Cl (HA), 100 nM CNP, 1 mM SNP, or combinations of HA+CNP and HA+SNP. ROS production was measured every 30 min for 6 h on a plate-reading spectrophotometer at 500 nm, with a final reading taken after 24 h. The data shown are means ± SEM, pooled from 4 to 6 independent experiments, each performed with 16 replicates (*n* = 4 to *n* = 6), and normalised to the 24 h control reading. **** *p* < 0.0001, significantly different from control.

**Figure 8 cells-10-00398-f008:**
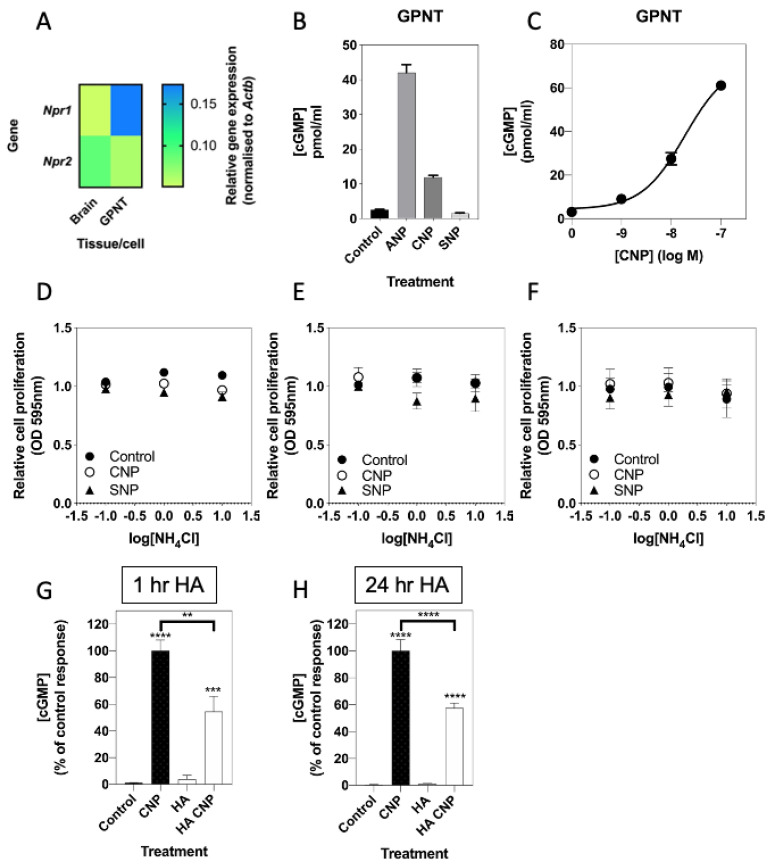
Effects of hyperammonaemia on the natriuretic peptide system in rat GPNT brain endothelial cells. (**A**) Multiplex RT-qPCR was performed on RNA extracted from rat brain tissue or GPNT cells. The data shown are mean relative gene expression, normalized to *Actb*, of 2 individual RNA extractions. (**B**) Total cGMP accumulation in GPNT cells treated with 0 or 100 nM ANP, CNP or 1 mM SNP for 1 h in physiological saline solution containing 1 mM IBMX, before assay. Data shown are means ± SEM of triplicate treatments. (**C**) Concentration-dependent effects of CNP on cGMP accumulation in C6 cells. Cells were treated with the indicated concentrations of CNP in PSS containing 1 mM IBMX for 1 h. Data shown are means ± SEM pooled from three independent experiments, each performed in triplicate. (**D**–**F**) Effects of hyperammonaemia and CNP on cell proliferation in GPNT cells. GPNT cells were treated with the indicated concentrations of NH_4_Cl in the absence or presence of 100 nM CNP or 1 nM SNP for (**D**) 24 h, (**E**) 48 h or (**F**) 72 h, before being fixed in 4% (*w*/*v*) paraformaldehyde and stained for crystal violet assay. The data shown are means ± SEM pooled from 3 independent experiments, each performed with 8 replicates. (**G**,**H**) Effects of hyperammonaemia on CNP-stimulated cGMP accumulation in GPNT cells. Cells were initially treated in media containing 1% (*v*/*v*) FCS and 0 or 10 mM NH_4_Cl (HA) for either (**G**) 1 h or (**H**) 24 h, prior to subsequent stimulation with 100 nM CNP in the presence of 1 mM IBMX for 1 h. The data shown are means ± SEM of five to seven independent stimulations, expressed as the percentage of the control response to CNP; ** *p* < 0.01, *** *p* < 0.001, **** *p* < 0.0001, significantly different from untreated (control), or from the control CNP response to CNP (brackets).

**Figure 9 cells-10-00398-f009:**
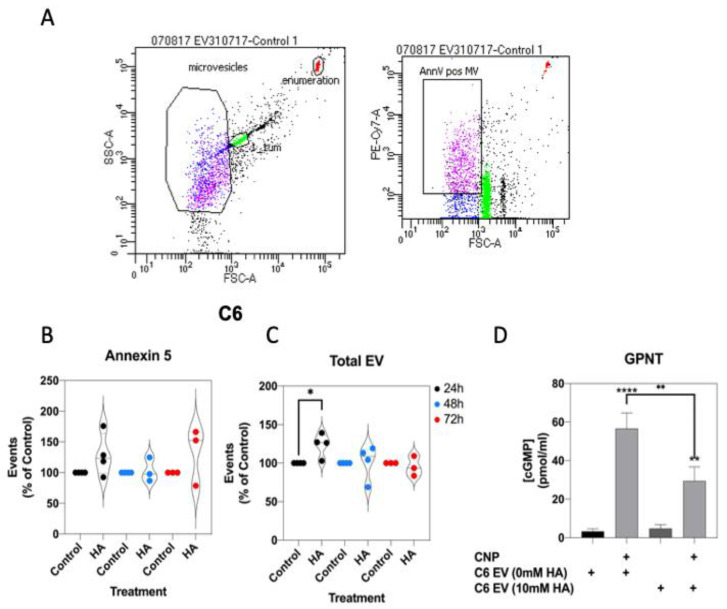
Effects of hyperammonaemia on extracellular vesicle production from C6 cells, and their functional effects on GPNT cells. (**A**) Extracellular vesicles were harvested from spent media from C6 cells treated with 0 or 10 mM NH_4_Cl for up to 72 h. These vesicles were stained with Cy7-labelled Annexin-5 and analysed by FACS. Data shown are representative FACS dot plots, showing gating parameters. (**B**,**C**) Comparison of Annexin-5-positive extracellular vesicles, and total extracellular vesicles from C6 cells treated with 0 or 10 mM NH_4_Cl. Data shown are medians (with 5 to 95% confidence intervals) of events, normalized to % of control treated cells, from 3 to 4 independent experiments. * *p* < 0.05, significantly different from control). (**D**) Effects of C6-derived extracellular vesicles on CNP-stimulated cGMP accumulation in GPNT cells. GPNT cells were treated with extracellular vesicles from control or NH_4_Cl-treated C6 cells, for 24 h, prior to subsequent stimulation with 100 nM CNP in the presence of 1 mM IBMX for 1 h. The data shown are means ± SEM from 6 to 9 independent stimulations. **** *p* < 0.0001, ** *p* < 0.01 significantly different from unstimulated; * *p* < 0.05, significantly different from CNP response in C6 EV (0 mM HA) cells.

## Data Availability

The data presented in this study are available on request from the corresponding author.

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
