# Peer review of "Sensitivity of the Natriuretic Peptide/cGMP System to Hyperammonaemia in Rat C6 Glioma Cells and GPNT Brain Endothelial Cells"

_cells, 2021, doi:10.3390/cells10020398_

Round 1
Reviewer 1 Report
I have a few minor comments concerning additional data in revision.
1.Measurement of ROS:
Measurement of ROS was determined by fluorometrically in ref. 40. Both excitation and emission wavelength should be described in the text.
2. Fig. 6: ROS production
Please show significant difference between control vs HA, and SNP vs HA+SNP in Fig. 6D.
Author Response
1) Thank you for pointing this out, and we apologise for not providing sufficient information in general for the ROS assay methodology. We have re-written the ROS assay section, accordingly (p4 Section 2.6).
2) Although hyperammonaemia in this series of experiments modestly increased ROS production similarly to the response seen in the CNP experiments (Fig.6B), this effect failed to attain significance, in light of the profound ROS response to SNP. In keeping with other data presentation, we have not annotated the figure with those comparisons that failed to attain significance.
Reviewer 2 Report
This study suggested that hyperammonemia—should not be abbreviated as HA—pre-treatment caused a significant inhibition in subsequent C-type natriuretic peptide (CNP)-stimulated cGMP accumulation in both C6 glioma cells and brain vascular endothelial cell (GPNT) cells (i.e., multiple components of the blood brain barrier), suggesting that blood brain barrier (BBB) could be impaired by hepatic encephalopathy. However, BBB function is not impaired in chronic hepatic encephalopathy (not fulminant liver failure). For example, in cerebrospinal fluid of patients with hepatic encephalopathy, elevated levels of total protein, inflammatory cells, and bile acids are usually not observed. Although these in vitro data demonstrated functional interaction between CNP signaling and hyperammonemia in the models of astrocytes and endothelial cells, the mechanisms of in vivo or clinical hepatic encephalopathy have not been fully discussed in the Introduction section. With regard to hepatic encephalopathy or ammonia toxicity, an important function of astrocytes is that of ammonia detoxification, whereby ammonia is converted to glutamine; increased levels of glutamine and alpha-ketoglutarate in cerebrospinal fluid are highly specific markers in patients with hepatic encephalopathy. It is also reported that cerebrospinal fluid lactate is increased in hepatic encephalopathy, suggesting that some metabolic derangement or dysfunction of the Krebs cycle is involved in the brain of hepatic encephalopathy.
Author Response
Thank you for your comments. We have removed ‘HA’ as an abbreviation in the text and replaced it with hyperammonaemia (although HA is retained as an abbreviation in the figures). We have also significantly reduced the emphasis on hepatic encephalopathy throughout the manuscript, and have broadened other mechanism of hyperammonaemia and neurological disorders that may affect cGMP signalling.
Reviewer 3 Report
I fully respect the authors’ efforts towards revising the manuscript which is now complemented with several new data. However, the new data are neutral to my critical opinion regarding the use of C6 to mimic the aspects of ammonia gliotoxicity typical for hyperammonemia or hepatic encephalopathy. From the authors’ response I conclude that in my review of the first version of their manuscript I have not not presented this point with enough precision. Please find below some clarifying comments:
- C6 cells as studied here present an undifferentiated phenotype, whereas in HE or HA in humans or experimental animals changes occur in differentiated astrocytes. Therefore, the effects of ammonia on C6 cells will not mimic astrocytic changes typical of HE. First, even undifferentiated astrocytes greatly differ in their responses to ammonia from mature (differentiated) astrocytes (Kala and Hertz, Neurochem Int. 2005 and references therein). Second, in a direct comparative study, C6 cells turned out to be more sensitive to ammonia-induced damage than rat (differentiated) astrocytes in primary culture (Haghighat et al., Neurochem Res 2000).
- C6 fundamentally differ from mature astrocytes in ammonia metabolism. The intensity and hence, the role of glutamine synthesis is the crucial difference. In astrocytes ammonia is primarily trapped by glutamate and converted to glutamine by glutamine synthase (GS), and under increased ammonia load, excessive accumulation of glutamine is an important trigger of mitochondrial dysfunction, oxidative/nitrosative stress etc. Glutamine synthesis in C6 cells is comparatively very inefficient (Portais et al., FEBS Letters, 1993). This fundamental difference between astrocytes and C6 cells also became apparent in the study of Galland et al (NCI, 2019), brought up by the authors in defense of their view. Fig. 3 in the paper by Galland demonstrates that glutamate uptake, glutamate transporter expression and GS activity in C6 cells are very poor as compared to astrocytes in primary culture. In their concluding comments, Galland et al state “Based on our data, although PC (primary cultures of astrocytes – reviewer’s footnote) is a more time-consuming and costly technique, its use as a reproductive model for astrocyte studies is undoubtedly preferential”.
All in all, without validation in cultured astrocytes, the effects of ammonia found by the authors in C6 cells cannot be considered as reflecting astrocytic status in HE.
Author Response
We appreciate the reviewer’s opinions on the suitability of C6 glioma cells as a model for hepatic encephalopathy. Therefore, we have significantly altered the emphasis of the manuscript, such that it no longer focuses on hepatic encephalopathy, and we have removed hepatic encephalopathy from the title and the keywords. Instead, the manuscript focuses on the ammonia sensitivity of CNP signalling in C6 glioma cells and GPNT cells. We no longer refer to C6 glioma cells as model astrocytes, describing them as cells ‘that have phenotypic features in common with astrocytes’ (p16, line 493). We provide further discussion of the limitations of C6 cells (lines 503-507) to acknowledge their differences in ammonia metabolism.
This reviewer previously criticised the manuscript for lack of originality. All data presented in this significantly revised manuscript are original, and no other publications describe the same experiments that we present. Our choice to use cell lines instead of primary rat cells is justified, given the absence of comprehensive, or convincing, data in previous publications relating to CNP and hyperammonaemia. Of the three previous publications on this topic, only two studies have investigated CNP function, and none of them use either C6 cells or GPNT cells. Furthermore, the concentrations of CNP used for these previous studies were beyond physiologically normal parameters for CNP (a locally produced peptide hormone), at some 100 times greater than the EC50 value for CNP. In light of the limitations of these earlier studies, it was prudent to perform more comprehensive studies of CNP signalling in an in vitro system, rather than utilise primary rat cells. We acknowledge the reviewer’s comments on our use of ‘undifferentiated’ C6 cells; this is an interesting point, and differentiating C6 cells towards a more astrocytic phenotype would be an interesting extension to our current study. However, it is confounded by the requirement to treat C6 cells with cAMP analogs, which in turn would alter PDE activity, rendering interpretation of our studies almost impossible.
All models of hepatic encephalopathy are flawed, whether they are cell lines, primary cultures, ex vivo brain slices, in vivo dietary induced HE, or in vivo bile-duct ligation surgery. Given the current crisis in rigour and reproducibility in science, it is reassuring that we have used an alternative system to those previously published, yet observed similar responses of CNP to hyperammonaemia.
Reviewer 4 Report
The experiments are well performed and the manuscript is well written. The major result of the manuscript is that ammonia reduces CNP-induced cGMP production in C6 glioma cells. This information was previously available in cortical astrocytes and RBE-4 cells. In addition, the characterization of the GPNT cells is described as far as cGMP signalling. Regarding the effect of ammonia, these cells are similar to C6 cells except their proliferation is not affected. Thus, there is some originality presented in the manuscript albeit very limited.
In addition, a hypothesis is presented according to which ammonia toxicity (e.g. hyperammonemia of hepatic origin) could be treated by increasing cGMP in glial cells elicited by CNP or CNP analogues. The results of the paper weaken this hypothesis as ammonia itself inhibits cGMP induced by CNP. Indeed, it is presented that CNP cannot counteract the toxic and anti-proliferative effects of ammonia on C6 cells and GPNT brain endothelial cells. In agreement with the lack of effect of CNP on ammonia-induced toxicity, ROS production elicited by ammonia was also not effected by CNP, which further limits the significance of the results.
A gene expressional study was performed to understand the mechanisms how ammonia affects C6 cells. It is not properly explained why the levels of astrocyte markers, such as GFAP and S100b are significantly lower in C6 ells than in brain tissue homogenates, which contains neurons, too. Most importantly, however, the mechanisms of ammonia actions cannot be explained by the detected gene expressional alterations. Moreover, an increased level of CNP receptors was found in response to ammonia, which is contradictory to the reduced cGMP levels. The brief explanation that it could be a compensatory mechanism is not fully satisfactory and altogether weakens the value of the paper.
Small remarks:
Ammonia is added to the cells by ammonium-chloride. It is claimed that ammonia is formed by metabolism from ammonium-chloride, which is not true, ammonia is formed simply by chemical equilibrium, which largely depends on the pH, which was not provided, and estimated ammonia levels were also not presented.
Figure 2 should indicate the concentration of ammonium-chloride. Is it molar or mM?
Only one housekeeping gene, beta-actin was used in the gene expressional study. The unaltered level of beta-actin in the given conditions was not demonstrated. Therefore, a combination of different housekeeping gene would be required.
Author Response
1) Thank you for these comments. We respectfully disagree that there is a lack of originality in our study. The previous studies that the reviewer refers to are neither comprehensive, nor convincing in demonstrating the effects of hyperammonaemia on CNP signalling. There are no other studies that report hyperammonaemic effects on CNP in C6 glioma cells or GPNT cells. We are reassured that some of our data reproduces findings seen in these previous other model systems, but we have extensively added to the understanding of ammonia sensitivity of CNP signalling.
2) We have significantly re-written the manuscript to change emphasis away from hepatic encephalopathy, towards a more comprehensive investigation of the ammonia sensitivity of CNP signalling in C6 glioma and GPNT cells. This is of relevance to several other neurological conditions in addition to HE.
3) We now mention the undifferentiated nature of C6 cells vs primary brain tissue as being a possible explanation for altered expression levels (line 500-501). Thank you for commenting on the receptor expression data – we realised the significance annotation for the Npr2 was incorrect, and there was no significant effect of HA on Npr2 expression. However, the significant increase in Npr3 is consistent with reduced cGMP signalling. Compensation happens in multiple systems, although not always effectively – nevertheless, further investigation is required to determine whether these observed changes in gene expression are functionally relevant to inhibited CNP signalling in the presence of hyperammonaemia.
Ammonia is added to the cells by ammonium-chloride. It is claimed that ammonia is formed by metabolism from ammonium-chloride, which is not true, ammonia is formed simply by chemical equilibrium, which largely depends on the pH, which was not provided, and estimated ammonia levels were also not presented. Thank you for this, we have removed reference to it.
Figure 2 should indicate the concentration of ammonium-chloride. Is it molar or mM? Thank you, we have added mM to the figure legend.
Only one housekeeping gene, beta-actin was used in the gene expressional study. The unaltered level of beta-actin in the given conditions was not demonstrated. Therefore, a combination of different housekeeping gene would be required. Both Actb and Rpl19 were used in the multiplex assay, and Actb was found to be a far more reliable housekeeping gene.
Reviewer 5 Report
I find that the present work has little relevance to be published in this journal.
The main concerns are:
- The cell models used do not have sufficient physiological relevance, since the results and consequences can be very different in the whole brain or at least brain slices, considering the inter-relationship between different cell types. Furthermore, they are not even primary crops. Cell function and responses can be very different from cells found in the brain in vivo.
- The ammonium concentration of 10 mM does not make any sense, it is never reached, even under pathological conditions.
- The results are scarce and are not very new or very relevant in the field of hyperammonemia and hepatic encephalopathy.
Finally, I wanted to indicate that the manuscript PDF is with the changes made visible, which makes it really difficult to read.
Author Response
- Many thanks for this observation. Cell lines are extremely useful tools in order to conduct fundamental experiments, but we are very aware that they are not the same as primary cultures, organ slices or whole animal experiments. However, it would have been morally and ethically questionable (and completely at odds with NC3Rs) to perform all of our studies in primary astrocytes and endothelial cells, without having first characterised the sensitivity of the natriuretic peptide system to hyperammonaemia in the C6 and GPNT cells. Whilst GPNT cells have not been extensively studied (making our data novel observations), since 1975 there have been nearly 800 published studies using C6 glioma cells as model astrocytes. In our manuscript, we make no claim whatsoever that our findings faithfully mimic the conditions found in primary astrocytes, and have been careful not to over-interpret our data. We are aware of literature that reports differences in phenotype between C6 cells and primary rat astrocytes, and our own expression data (included in this manuscript) reveals some differences in gene expression between C6 cells and primary rat brain tissue. Where we have performed similar studies in C6 cells to those reported in primary rat astrocytes, we have essentially confirmed the effects of hyperammonaemia on CNP signalling (although we have used nM concentrations of CNP, compared to others using micromolar) – this, in itself, is reassuring that in a key phenotypic property for our studies, C6 cells can faithfully reproduce the response to that seen in primary astrocytes.
- As the reviewer is aware, we used a range of NH4Cl concentrations (1mM, 5mM, 10mM), and focussed on 10mM as it provided the most robust response on cGMP in both cell lines. We are aware of the literature that uses 5mM NH4Cl as an upper concentration – however, there are several previous studies that have all used 10mM NH4Cl, and we have now cited these in the Methods section (p3, Materials) (for info: Neary et al, 1990; Blancoa et al, 2013; Hilgier et al, 2010; Yang et al, 2011; Laemmle et al, 2016).
- We respectfully disagree with the reviewer. The vast majority of the data we present are novel, and use either novel experimental designs or different cell lines. We do include data that confirm the findings of previously published studies – in the current climate where rigour and reproducibility in science is being strongly scrutinised, we believe our ability to reproduce some of these findings to be a positive attribute. Nevertheless, the very few (there are 2 (TWO) publications on CNP and ammonia on PubMed) previously published studies (on ammonia inhibition of CNP signalling) did not go in to as much detail as we do in the current manuscript, and these studies employed micromolar concentrations of CNP (which are arguably not physiological). With regards to the criticism about relevance to hepatic encephalopathy, we have significantly re-written the manuscript, and considerably reduced the emphasis on HE, highlighting that both hyperammonaemia and a reduction in cGMP concentrations are features of numerous other neurological complications that are independent of liver function.
We apologise for the PDF version – track changes were switched off prior to uploading the revised manuscript, so we were disappointed to see the PDF version still including these annotations.
Reviewer 6 Report
The paper from Regan et al. investigates the impact of high ammonia on the astrocytic and endothelial natriuretic peptide system, focusing on how ammonia impair the Type-C natriuretic peptide-mediated increase of cyclic GMP. The paper is interesting as it characterizes both astrocytes and endothelial cells' response to different natriuretic peptides and investigates mechanisms of action that could potentially be affected in conditions of high ammonia. However, some points require attention.
Major points
- The paper is not on an in vitro model of hepatic encephalopathy since it does not show that ammonia impairs cGMP (control vs. HA), nor it assessed the potential of CNP to reverse such effects. This study only evaluates the pharmacological and molecular mechanisms by which ammonia impairs the exogenous CNP-stimulated cGMP release. Therefore, it should not mention hepatic encephalopathy on either the title or the aim. Also, since no evidence in this paper merits the discussion of CNP as a potential drug for HE, that should be removed from the introduction and discussion.
- Figure 5) Please explain why the authors compare the efflux results (figure A) from CNP vs. CNP+HA groups (with cGMP efflux changes) with the gene expressions from the Control vs. HA groups (without cGMP efflux changes?) The gene expression should be done for the same groups with the differences in efflux to be comparable.
- The proper control group for ammonium chloride (sodium chloride at equivalent concentrations) should be added, so differences due to osmolality changes are excluded.
Minor points
- In the introduction, the authors mention that cGMP impairments are associated with HE. Please, elaborate further on this matter (which symptoms and what are the known mechanisms).
- Figure 1) The x-axis for figure B should be in pmol/ml
- Please clarify the timeline between the increased ROS at 24h and the Hmox gene changes at 48.
- The discussion states that there is desensitization of the GC-B receptor by CNP exposure. Please, clarify where this finding is and add it to the results section.
- Please, include in the discussion the limitations of this paper related to the lack of proof that high ammonia alone impairs the CNP-cGMP pathway. Explain why the differences were only seen when CNP was added and what that means for the clinical context.
- There are typos in the discussion section that need to be corrected.
Author Response
Thank you for all your comments on our manuscript. We agree that it is too contentious to present our in vitro investigations in C6 and GPNT cells as a model of hepatic encephalopathy, and have altered the title, and the emphasis of the entire manuscript, accordingly.
Figure 5) Please explain why the authors compare the efflux results (figure A) from CNP vs. CNP+HA groups (with cGMP efflux changes) with the gene expressions from the Control vs. HA groups (without cGMP efflux changes?) The gene expression should be done for the same groups with the differences in efflux to be comparable. We completely agree with the reviewer. Ideally, we would have had the time, and resources, to perform the multiplex RT-qPCR assays on all samples from all time points. Unfortunately, we were unable to do so, and made the decision to examine the effects of hyperammonaemia on the expression of PDE and cGMP pathway genes, arguing that the robust inhibitory effect on CNP-stimulated cGMP efflux could be mediated via changes in gene expression. We agree that linking the gene expression data with the spent media experimental samples (from which cGMP levels were unaffected, as the cells were not stimulated with CNP), is inappropriate. We have, therefore, separated out these figures, and altered the text accordingly to remove the link between these two aspects of the experiment. Additionally, we have added text (lines 605-610) to the discussion to clarify this. We are not able to perform any further RT-qPCR analysis as our laboratory has been closed since 18th March 2020, and we are not able to currently gain access.
The proper control group for ammonium chloride (sodium chloride at equivalent concentrations) should be added, so differences due to osmolality changes are excluded. Thank you for this - we have not performed this control. Whilst we acknowledge that NaCl is used for osmolarity corrections in some studies, several other studies do not correct for such changes (refs: Buzańska et al, 2000; Konopacka et al, 2008; Dai et al, 2013; Santos et al, 2015; Warskulat et al, 2001).
Minor points
In the introduction, the authors mention that cGMP impairments are associated with HE. Please, elaborate further on this matter (which symptoms and what are the known mechanisms). We have added examples of this to the introduction (oedema, cognitive dysfunction), and also describe altered cGMP signalling in other neurological disorders that are independent of hepatic function (lines 87-94, 477-488).
Figure 1) The x-axis for figure B should be in pmol/ml – Thank you, we have altered the axis (and the text) (lines 218-222)
Please clarify the timeline between the increased ROS at 24h and the Hmox gene changes at 48. Upregulation of HMOX would be expected as a protective mechanism subsequent to increased ROX production. In our current study, the absence of gene expression data after 24h of hyperammonaemia make it difficult to establish a direct link between the modest effect on ROS, and the enhanced Hmox expression. As we have mentioned above, we would have preferred to analyse all time points and treatments, for the gene expression studies.
The discussion states that there is desensitization of the GC-B receptor by CNP exposure. Please, clarify where this finding is and add it to the results section. Thank you for pointing this out. We have amended Figs3 and 4 to include the rates of accumulation linear regression data, and describe these findings in the results and discussion (lines 298-300, 316-318, 560-582).
Please, include in the discussion the limitations of this paper related to the lack of proof that high ammonia alone impairs the CNP-cGMP pathway. Explain why the differences were only seen when CNP was added and what that means for the clinical context. Many apologies, but we are not sure how to address this comment, especially as the reviewer has already asked us to remove any insinuation that our studies are reflective of CNP signalling under conditions of hepatic encephalopathy. We already make the point that gene expression studies in the presence of CNP would be important to perform.
There are typos in the discussion section that need to be corrected. Thank you, we hope we have altered all of these.
Round 2
Reviewer 3 Report
Appreciably, in the revised manuscript the authors distanced themselves from interpreting their results as pertaining to the status of astrocytes in hepatic encephalopathy, focusing on how ammonia affects CNP signaling in two cell lines that resemble in a considerable degree their native CNS counterparts. Further, the authors accentuated the novelty of the study, which consists in more detailed description of the CNP signaling pathway than reported in previous studies by other authors. In this way the authors convinced the referee about the merits of their study. Congratulations.
Reviewer 4 Report
The authors responded to the critiques well and supplemented their data with additional analysis.
Reviewer 5 Report
I appreciate the clarifications of the authors.
The work has gained news by adding the extracellular vesicles.
The work is quite complete.
Although I still have doubts about the model, it is true that it is a first step to study the role of CNP in hyperammonemia and thus the use of animals can be reduced, from an ethical point of view.
Although I am aware of the publication of previous work with these ammonium concentrations, the problem of ammonium concentration is not solved either because it is still not very relevant at a physiological level. At best it would be for acute ammonium poisoning, but not under conditions of chronic hyperammonemia. This can be commented in the manuscript, when the hyperammonemia model used is presented.
The authors could comment on the fact that astrocytes have a large amount of glutamine synthetase that reduces ammonium concentration rapidly and the possible role of glutamine synthetase on the effects of hyperammonemia on cGMP and CNP signalling.
This manuscript is a resubmission of an earlier submission. The following is a list of the peer review reports and author responses from that submission.
Round 1
Reviewer 1 Report
In this manuscript “Molecular and pharmacological consequences for the natriuretic peptide system during hyperammonemia in rat C6 glioma cells: Implications for hepatic encephalopathy” Cells-857547, Regan investigated the effect of hyperammonemia on CNP-stimulated cGMP production in C6 glioma cells. The authors reported that hyperammonemia suppressed CNP-stimulated cGMP production without any effect on cell viability. They also revealed that several gene expressions were upregulated such as Pde5a, Pdde11a, Mrp4, Npr2 and Npr3 under heperammonemic condition. There are some interesting aspects of this manuscript, however, there are some critical issues should be addressed as described below.
Major:
- This is a similar study previously carried out by other group concerning the CNP-stimulated cGMP production under hyperammonemic condition in brain (ref 16-18).
In addition, I have one serious caveat, however: the use of C6 glioma cells as an astrocyte model. There are some characteristic differences between C6 glioma and primary astrocytes as the authors described in Fig. 1A. Therefore, I strongly suggest that the manuscript would be considerably strengthened by verifying at least some of these results in primary astrocyte culture.
- The authors concluded that CNP-stimulated cGMP production is mediated via GC-B/Npr2 pathway. However, there was no direct evidence in this manuscript. It would be appropriate to check the result in the presence of GC-B antagonist in this study.
- The discussion lacks a mechanistic approach. How did hyperammonia upregulate several gene expressions?
- The CNP-stimulated cGMP accumulation was inhibited under hyperammonemic condition for 1 h (Fig. 3A). Were there any changes in gene expression under hyperammonemia for 1 and 24 h?
Minor:
- How did the authors get rat brain tissue in Fig. 1A? It was not mentioned in Materials and Methods.
- Are there any morphological changes in C6 glioma under hyperammonemic conditions?
- Actual measurement value of cGMP should be listed in Figs. 2-5. These information is valuable for general readers.
I hope these comments will be helpful.
Reviewer 2 Report
In hepatic encephalopathy (HE), hyperammonemia is one of the major contributing factors to the neurological disturbances that affect astrocytes of the central nervous system. C-type natriuretic peptide (CNP), but not atrial natriuretic peptide, acts via the guanylyl cyclase receptor-B (GC-B/Npr2) to increase the formation of cGMP in its target tissues. Astrocytes—the most common numerous cell type of the central nervous system—perform a variety of roles including ammonia detoxification. CNP, via the GC-B receptor, has effects on brain function, neuroprotection, and blood-brain-barrier permeability. In this study by Regan et al., the effects of hyperammonaemia on CNP signaling in the well-characterised rat C6 glioma cell line—the model of astrocyte—to establish how the astrocyte natriuretic peptide system might be affected in patients with HE. There was a significant, concentration-dependent inhibition in cell proliferation after 72h of hyperammonaemic conditions; the cell proliferation was not prevented by the presence of CNP. Pre-treatment in hyperammonaemia conditions caused a dose-dependent inhibition of CNP-stimulated cGMP accumulation. This in vitro study suggests that cGMP generation mediated via membrane guanylyl cyclases (specifically GC-B/Npr2) is sensitive to hyperammonaemia, by a mechanism that does not appear to affect cGMP generation through soluble guanylyl cyclase activation (i.e., through NO).
Major comments
- The concentrations of NH4 up to 10 mM (i.e., hyperammonaemia) seems too high compared with those in the clinical situation. Please provide the rationale for this experimental setting. The normal range of blood ammonia is approximately 18–47 μmol/L. Even in patients with liver cirrhosis or hepatic encephalopathy, blood ammonia levels are usually modestly increased (approximately 200 μmol/L) (Aldridge DR, et al., J Clin Exp Hepatol 2015;5:S7-S20).
- Transcripts encoding phosphodiesterase enzymes (Pde5a and Pde11a) and other cGMP-associated genes (Mrp4, Npr2, and Npr3) were increased (i.e., up-regulation in transcripts) under the conditions of hyperammonaemia (ANOVA followed by Turkey’s or Dunnett’s multiple comparison tests). Although one study [Ref. 51] showed that ammonia increased both mRNA and its protein in rat astrocytes and human cerebral cortex from patients with hepatic encephalopathy, it is not clear that up-regulated transcripts could result in synthesis of corresponding proteins, where energy production system is impaired by hyperammonaemia (e.g., increased brain lactate [Butterworth, Drugs 2019;79:S17-S21]).
Minor points
- These words such as hyperammonaemia (HA) and central nervous system (CNS) do not need to be abbreviated.
- What is “A” at line 395.
Reviewer 3 Report
The authors address the role of the natriuretic peptide system in ammonia-affected astrocytes. In terms of biochemical techniques the study is performed correctly. However, two aspects of this work play down its significance: a) limited relevance of the cell population used, b) lack of essentially novel information to be derived from the study.
a) The study was performed on C6 cells - an astroglioma cell line whose phenotype is very remote from astrocytes. The metabolic processes in which C6 cells differ from astrocytes (native or cultured) directly pertain to the role of astrocytes in brain physiology and pathology. These include among others the mechanism of glutamate transport (predominantly by xc- transporter mediated Glu-Cys exchange, not GLAST/GLT1, Cho and Bannai, J Neurochem 1990), or energy metabolism (excessive lactate production vs consumption, Bouzier et al., JBC 1998). At the gene expression level, substantial differences have been found regarding expression of purinergic enzymes and transporters (Parkinson et al., J Neurosci Res 2006 Sep;84(4):801-8. Therefore, the results of the present study cannot be safely considered as representing astrocytic response. Down this valley, the absence of increase by ammonia of SNP-evoked NO synthesis in ammonia-treated C6 cells (Fig. 4) is at variance with what may be predicted for ammonia-treated astrocytes, as ammonia under conditions similar to the present study invariably increases iNOS expression and activity in cultured astrocytes (Schliess et al., FASEB J. 2002 16(7), 739-741; Sinke et al., J Neurochem. 2008, 106, 2302-2311; Zielinska et al., J Neurochem, 2015, 135, 1272-1281, and many other references). Further confirmation of observations made here cultured astrocytes is always strongly recommended. However, this recommendation does not hold for the present study because:
b) Important observations on the role of CNP in ammonia-treated astrocytes, exceeding in number and significance those reported here, have already been made in cultured astrocytes, or in in vivo/ex vivo systems in which experimental conditions are designed to distinguish astrocytic response. Some the previous studies, showing responses in cultured astrocytes and endothelial cells, have been duly quoted by the authors (refs. 16-18). Other, more recent observations include demonstration of the modulatory role of CNP/PKG pathway in ammonia-induced astrocytic swelling and brain edema (Konopacka et al., J Neurochem, 2009, 109, 246–251) and of CNP as an attenuator of oxidative stress in ammonia-exposed astrocytes (Skowronska et al., J Neurochem. 2010 115, 1068-1076).
Relatively minor point:
The term “hyperammonemia” denotes increased blood ammonia in vivo, It should therefore be replaced with “ammonia exposure”.